# GPR41 Regulates the Proliferation of BRECs via the PIK3-AKT-mTOR Pathway

**DOI:** 10.3390/ijms24044203

**Published:** 2023-02-20

**Authors:** Zitong Meng, Dejin Tan, Zhiqiang Cheng, Maocheng Jiang, Kang Zhan

**Affiliations:** 1Institute of Animal Culture Collection and Application, College of Animal Science and Technology, Yangzhou University, Yangzhou 225009, China; 2Institutes of Agricultural Science and Technology Development, Yangzhou University, Yangzhou 225009, China; 3Joint International Research Laboratory of Agriculture and Agri-Product Safety, The Ministry of Education of China, Yangzhou University, Yangzhou 225009, China

**Keywords:** GPR41, BRECs, proliferation, PIK3-AKT-mTOR

## Abstract

Short-chain fatty acids (SCFAs) play a pivotal role in regulating the proliferation and development of bovine rumen epithelial cells (BRECs). G protein-coupled receptor 41 (GPR41) is involved in the signal transduction in BRECs as a receptor for SCFAs. Nevertheless, the impact of GPR41 on the proliferation of BRECs has not been reported. The results of this research showed that the knockdown of GPR41 (GRP41KD) decreased BRECs proliferation compared with the wild-type BRECs (WT) (*p* < 0.001). The RNA sequencing (RNA-seq) analysis showed that the gene expression profiles differed between WT and GPR41KD BRECs, with the major differential genes enriched in phosphatidylinositol 3-kinase (PIK3) signaling, cell cycle, and amino acid transport pathways (*p* < 0.05). The transcriptome data were further validated by Western blot and qRT-PCR. It was evident that the GPR41KD BRECs downregulated the level of the PIK3-Protein kinase B (AKT)-mammalian target of the rapamycin (mTOR) signaling pathway core genes, such as PIK3, AKT, eukaryotic translation initiation factor 4E binding protein 1 (4EBP1) and mTOR contrasted with the WT cells (*p* < 0.01). Furthermore, the GPR41KD BRECs downregulated the level of Cyclin D2 *p* < 0.001) and Cyclin E2 (*p* < 0.05) compared with the WT cells. Therefore, it was proposed that GPR41 may affect the proliferation of BRECs by mediating the PIK3-AKT-mTOR signaling pathway.

## 1. Introduction

The rumen of a ruminant is a unique fermenter capable of obtaining nutrients from indigestible plant matter to provide dairy products for humans. When a calf is born, the rumen is not fully developed, and its volume is about 30% of the total volume of the four stomachs. The rumen volume rises to 70% of the total volume upon the feeding of starter and forage-based diets, and from this stage, the weaned calves rely primarily on the rumen epithelium to absorb SCFAs for energy [1]. During this, the rumen epithelium undergoes significant changes [2,3]. Concurrently, the rumen epithelium also plays an important effect in immunity and barrier functions, and the perfection of its morphology and function is essential for ensuring the health of ruminants. Therefore, understanding the growth mechanisms of BRECs is crucial in developing strategies to increase ruminant productivity.

Numerous studies have shown that these physical and functional changes in the rumen epithelium arise from SCFAs [4]. The results of perfusion experiments identified SCFAs as key nutrients that stimulate the morphological development of rumen epithelial cells and papillae, among which butyric acid played a particularly significant role [1,5]. In addition, in vivo experiments confirmed that butyric acid perfusion promotes the rumen epithelial cell cycle from the G0/G1 phase to the S phase, with cells able to synthesize DNA in the S phase, thereby promoting mitosis and proliferation, which in turn affects the growth and development of rumen papillae [6]. However, our previous findings suggest that butyrate inhibits the proliferation of BRECs in vitro by reducing the levels of genes and proteins associated with the G0/G1 checkpoints in the cell cycle [7]. This phenomenon led us to contemplate whether SCFAs could promote the growth of the rumen epithelium through the molecular pathways.

The GPR41 is expressed on the epithelial cells of the rumen of dairy cows, which is a specific receptor for SCFAs. Recent research has shown that GPR41 is linked with histone acetylation and may be pertinent to the regulation of acetylation-related cellular processes, including proliferation, apoptosis, and the cell cycle [8,9,10]. A previous study of ours unexpectedly found that the GPR41KD inhibited the proliferation of BRECs, but the exact mechanism was not clear. The PIK3-AKT-mTOR pathway is an important intracellular signaling pathway that regulates the cell cycle. Therefore, it is directly related to cell development, proliferation, cancer and lifespan. In addition, further studies have shown that the PIK3-AKT-mTOR signaling pathway is positively correlated with cell proliferation [11,12,13]. In addition to this, our previous transcriptome data analysis showed that differential genes were enriched in the PIK3-AKT-mTOR signaling pathway. Hence, we speculate that GPR41 may regulate the proliferation of BRECs through the PIK3-AKT-mTOR signaling pathway. To our knowledge, this is the first study to reveal the relationship between GPR41 and the PIK3-AKT-mTOR signaling pathway. In this study, we investigated the effect of GPR41 on the proliferation of BRECs through the PIK3-AKT-mTOR signaling pathway with a view to providing a new basis for cow development.

## 2. Results

### 2.1. Proliferative Activity Analysis

In the wild-type BRECs (WT) group, cell proliferation increased steadily from day one to day four. Contrasted to the WT, the proliferation of GPR41KD BRECs was reduced on the first day of culture (*p* = 0.009), and their differences were greatest on the fourth day (*p* < 0.001, Figure 1).

### 2.2. Transcriptome Analysis

RNA sequencing (RNA-seq) analysis revealed that the differential genes in GPR41KD BRECs were enriched in the PIK3-AKT-mTOR pathways in the cancer signaling pathways compared to WT (Figure 2). In addition, our transcriptome profiling data showed that the genes related to amino acid transport, such as *SLC1A1*, *SLC1A5*, *SLC7A1*, *SLC7A7*, *SLC7A11*, and *SLC38A1*, were downregulated (*p* < 0.001). Furthermore, the *SLC2A5* and *SLC5A3* genes associated with Glucose transport were suppressed (*p* < 0.001), whereas there was no variation in the level of the *SLC2A11* and *SLC5A9* genes. The PIK3-AKT-mTOR signaling pathway genes, such as *PIK3CA*, *PIK3CB*, *PIK3CG*, *PIK3C2A*, *PIK3C2B*, *mTOR*, *RPS6KA2*, *KRAS*, and *ERK4*, are downregulated when compared with that in the WT BRECs (*p* < 0.001). Consistent with the amino acid transport, the cell cycle genes are simultaneously reduced (*p* < 0.001). The genes involved in Mitochondrial transport, including the *SLC25A15*, *SLC25A37*, and *SLC25A40* genes, were decreased (*p* < 0.001, Table 1).

### 2.3. Differences in Genes Expression

To further validate the RNA-seq data, the level of genes appertained in the PIK3-AKT-mTOR and cell cycle signaling pathway were analyzed by qRT-PCR. Contrasted to the WT cells, the mRNA levels of the key genes involved in the PIK3-AKT-mTOR signaling pathways, such as *PIK3* (*p* = 0.01), *AKT* (*p* = 0.01), *mTOR* (*p* < 0.001), eukaryotic translation initiation factor 4E binding protein 1 (*4EBP1*) (*p* = 0.008), and eukaryotic initiation factor 4E (*EIF4E*) (*p* = 0.02), were suppressed in the GPR41KD BRECs. However, the expression of ribosomal protein S6 kinase 1 (*S6K1*) mRNA was not profoundly altered (*p* = 0.29). On the other hand, there was a tendency to reduce Cyclin E2(*CCNE2*) and Cyclin-dependent kinases 2 (*CDK2*). In addition, the GPR41KD BRECs decreased the mRNA expressions of *p21^Cip1^* (*p* = 0.02) and the Cyclin D2 (*CCND2*) (*p* < 0.001) genes related to the cell cycle contrasted with WT, whilst there were no changes in the mRNA expression of *p27^Kip1^*, Cyclin E1(*CCNE1*), Cyclin- dependent kinases 4 (*CDK4*), and Cyclin D3 (*CCND3*) between the WT and GPR41KD treatment (Figure 3).

### 2.4. Differences in Proteins Expression

To validate the transcriptome analysis data and further confirm the effect of GPR41KD on the key proteins in the PIK3-AKT-mTOR signaling pathway, the protein level of the PIK3-AKT-mTOR pathway-linked genes was determined in the WT BRECs and GPR41KD BRECs by Western blot analysis (Figure 4). The level of PIK3 (*p* = 0.0136) and phosphorylated (p)-S6K1 (*p* = 0.0174) protein showed a slight downward trend by GPR41KD compared with the WT BRECs, whilst there was no variation in the level of 4EBP1 and p-AKT protein between the WT and GPR41KD treatment (Figure 4). Compared with the WT group, the protein abundance of AKT (*p* = 0.0011), S6K1 (*p* = 0.0092), and mTOR (*p* = 0.0051) were lower in response to the GPR41KD treatment (Figure 4). The relative protein abundances of p-mTOR (*p* = 0.0008) and p-4EBP1 (*p* = 0.0007) were lower in GPR41KD compared with the WT group (Figure 4).

## 3. Discussion

The rumen epithelium wields a significant function in the absorption, transport, and metabolism of SCFAs [14,15]. In a previous study, we found that GPR41 as a receptor for SCFAs may provide a molecular link between the diet and immune response in ruminants by establishing a GPR41KD BRECs cell line [16]. Interestingly, we unexpectedly discovered that GPR41KD had a suppressed impact on the proliferation of BRECs [17]. In addition, we have interpreted the cell proliferation inhibition caused by GPR41KD from a cell cycle perspective. In view of the above findings, we speculate that other pathways may exist for GPR41KD to inhibit the proliferation of BRECs.

The RNA-seq has become an essential tool for analyzing differential gene expression at the whole transcriptome level and studying differential mRNA splicing [18]. Therefore, we investigated the impact of GPR41KD on the mRNA level of proliferation-related genes by RNA-seq. Significant differences were found in the proliferation-related gene expression after GPR41KD compared to WT cells; the differential genes were enriched in the Amino acid transport, cell cycle, and PIK3-AKT-mTOR pathway. These data suggest that the gene expression differences are associated with *GPR41*. Consequently, we studied the regulatory impacts of *GPR41* on BRECs proliferation.

The AKT or PKB is a serine/threonine kinase with well-characterized roles in several basic cellular processes, including transcription, development, proliferation, and protein synthesis [19]. Factors such as the receptor tyrosine kinases, G protein-coupled receptors, and other kinases can activate AKT cascade signaling and induce the stimulation of phosphatidylinositol trisphosphate (3,4,5) phosphatidylinositol (PIP3) production [20]. Wang et al. found that Pleomorphic adenoma gene 1(*PLAG1*) up-regulated the p-PIK3 and p-AKT proteins and decreased the downstream gene Forkhead Box O3 (*FOXO3)*, thereby promoting the proliferation of primary bovine myoblasts [21]. This finding suggests that AKT positively regulates cell proliferation. Zhang et al. found that Myocyte Enhancer Factor 2D (*circMEF2D)* lowered the proliferation of bovine myoblasts by inhibiting the IGF2-PIK3-AKT pathway-related genes (*PTEN*, *PDK1*, *FOXO1*, and *AKT*) [22]. This is consistent with our experimental results. The Western blot of the present research indicated that AKT expression was diminished, and cell proliferation was reduced after GPR41KD. Therefore, the authors postulate that GPR41 may mediate BRECs proliferation through the AKT signaling pathway.

The AKT is a key mediator of the PIK3 pathway function [23,24]. The family of lipid kinases, termed PIK3, has been found to play an influential signaling role in many cellular processes and is a pivotal regulator mediating cell survival, proliferation, and differentiation [25,26]. Song et al. also confirmed that miR-483 increased bovine myoblast proliferation through the IGF1-PIK3-AKT pathway and downregulated the expression of key PIK3-AKT pathway proteins (IRS1, PIK3, PDK1, and AKT) [27]. Wang et al. confirmed that the overexpression of *circTTN* significantly upregulates the levels of *IGF2*, *IRS1*, *PIK3*, *PDK1*, and *AKT*, but the knockdown of *circTTN* diminished these levels of genes [28]. These studies show that the PIK3 signaling pathway is a positive effector of cell proliferation, and the results of this study also confirmed this assertion. The experimental results of this study showed that the gene level of the PIK3-related pathway was significantly suppressed after GPR41KD. Therefore, the authors presume that PIK3 wields a critical function in the GPR41-mediated proliferation of BRECs.

The mTOR signaling pathway is a central factor affecting cell growth [29]. As a cellular nutrient sensing and energy regulator, mTOR can integrate various stimuli and signaling networks to promote anabolism under nutrient-rich conditions, while blocking intracellular catabolism, including inhibiting autophagy, thereby promoting cell growth and proliferation [30,31]. Zhang et al. found that Annexin A2 (*AnxA2)* up-regulated *PIP3* expression, promotes mTOR phosphorylation and enhances the expression of *SREBP-1c* and *CCND1*, thereby promoting the proliferation of bovine mammary epithelial cells (BMECs) [32]. Li et al. also confirmed that Twinfilin 1 (*TWF1)* controls the mTOR and downstream *SREBP-1c* and *CCND1* signaling molecules to improve BMECs proliferation [33]. Our recent research findings are compatible with the former reports that the reduced level of genes and proteins associated with the mTOR pathway inhibits the proliferation of BRECs, which is directly associated with the knockdown of *GPR41*. GPR41KD reduced the level of pivotal proteins of the PIK3-AKT-mTOR signaling pathway, which explains the suppressed cell proliferation. This is consistent with the anterior results that observed a positive correlation between the PIK3-AKT-mTOR signaling pathway and cell proliferation [11,12,13].

The mTOR signaling controls cell growth and proliferation, primarily through the downstream targets 4EBP1 and S6K1, integrating various types of signaling inputs including energy, growth factors, and amino acids [34]. Both 4EBP1 and S6K1 are proteins that regulate translation initiation. These proteins are normally the principal controlled nub of the cellular translation regulation machinery [35]. The protein S6K1 acts as a downstream target of mTOR and can be activated through the phosphorylation of mTOR, stimulating ribosome biosynthesis [36,37,38]. The findings of the present research show that GPR41KD inhibits the phosphorylation of mTOR, which in turn affects the activation of 4EBP1 and S6K1. Previous studies have shown that the downregulation of S6K1 and 4EBP1 expression decreases epithelial cell renewal and proliferation [38,39]. The results suggest that GPR41KD causes the downregulation of S6K1, p-S6K1, 4EBP1, and p-4EBP1, which in turn decreased BRECs proliferation in the current study.

Proliferation often leads to an increased demand for protein by the cell. Hence, an adequate supply of amino acids as a substrate for protein synthesis is vital for the cell to maintain its proliferation drive [18]. The upregulation of *SCL1A1* and *SLC1A5* expression has been shown to increase glutamine’s cellular uptake, thereby promoting proliferation [40,41]. Wang demonstrated that the translational knockdown of *SLC7A1* mRNA leads to developmental delay and functional abnormalities in sheep fetuses [42]. Furthermore, *SIc38a2* senses amino acid availability and then transmits them to the intracellular signaling pathways. In addition, regulates protein synthesis, cell proliferation, and apoptosis via mTOR and the general control of nonderepressible 2 (GCN2) pathways [43]. In the present study, GPR41KD resulted in the downregulation of the mRNA expression of amino acid transport-related genes. Therefore, this suggests that GPR41KD may inhibit proliferation by mediating the downregulation of amino acid transport genes, thereby reducing substrates for intracellular protein synthesis.

The regulatory network of the eukaryotic cell cycle is highly conserved [44]. The discovery of cell cycle proteins and Cyclin-dependent kinases (CDK), the elucidation of ubiquitin ligase regulatory pathways, and the characterization of transcriptional control and the checkpoint signaling mechanisms have exposed universal cell cycle controlled tenets common to eukaryotes, and these components also drive cell proliferation [45,46]. *CCND2* is a pivotal cell cycle effector and a pivotal member of the D2-CDK4 (DC) complex of cell cycle regulators [47]. The *CCND2* and *CDK4* gene families regulate the cell cycle by controlling the G1-S phase transition, which is integral for cell proliferation and development [48]. The post-translational modification of *CCND2* phosphorylation decreases the binding of mTOR downstream of the PIK3/AKT target to the transcription factor *E2F*, causing cells to exit the G1 phase of the cell cycle [49]. Similarly to *CDK4*, the phosphorylation of pocket proteins by *Cyclin D-CDK6* may release them from the *E2F* transcription factor, which in turn regulates the course of the G1 phase. In addition, Zhang revealed that *CDK6* and *CDC25A* are cell cycle regulators during the G1 to S transition by overexpressing *CDK6* or *CDC25A* alone [50]. It was shown that BCL2 increases G0 and defers the G0 to the S transition, and that its cell cycle deferral effect is selective for cell cycle re-entry from G0 onwards [51]. In the current research, GPR41KD reduced the expression of mRNA levels of *CCND2*, *CDK6*, *CDC25A, BCL2*, and other cell cycle proteins, compared to WT. This connotes that GPR41KD may mediate cell proliferation inhibition by affecting the level of cell cycle proteins, CDK, and other ingredients of the nub cell cycle machinery.

In the rumen epithelium, cells require adequate nutrients to maintain continuous cell growth and proliferation. Cell growth demands adequate carbon, nitrogen, and free energy to provide the synthesis of proteins, lipids, and nucleic acids required for proliferation. Nutrient uptake is key to supporting the rate of macromolecular synthesis that satisfies growth, which is one of the keys to cell proliferation. GPR41KD reduces the level of genes associated to amino acid and glucose transport, which somewhat decreases nutrient uptake and cell proliferation. The inadequate nutrient uptake by cells slows the synthesis rate of the macromolecules required for the cell cycle, which further affects the level of cell cycle-linked genes and proteins and may mediate the containment of cell proliferation.

Furthermore, PIK3-AKT-mTOR is an extremely conservative signal transduction pathway associated with growth and proliferation in cells and wields numerous other functions in glucose and amino acid metabolism. Previous findings denote that the PIK3-AKT-mTOR signaling pathway positively correlates with cell proliferation. In conclusion, GPR41KD restrained the transport of glucose and amino acids by rumen epithelial cells, which resulted in insufficient nutrient uptake by the cells. The nutrient deficiency in rumen epithelial cells further affected the level of pivotal proteins of the PIK3-AKT-mTOR signaling pathway, leading to suppressive cell proliferation.

## 4. Materials and Methods

### 4.1. Cell Culture

The Institute for Animal Cultural Collection and Application of Yangzhou University (Yangzhou, China) supplied the cells, including WT and GPR41KD BRECs [13]. All of the experimental procedures appertaining dairy cows were in compliance with the guidelines of the Institutional Animal Care and Use Committee of Yangzhou University (SYXK (Su) IACUC 2012-0029). The WT and GPR41KD BRECs were seeded in DMEM/F12 medium at 37 °C and 5% CO_2_.

### 4.2. Cell Proliferation

Cell proliferation assays were performed using the Cell counting plate (Invitrogen, Shanghai, China). The WT and GPR41KD BRECs were cultured into 24-well plates at 50,000 cells per well. The cells were counted in triplicate wells from d 1 to 4, and the growth curves of the mean cell numbers were plotted.

### 4.3. RNA-Seq and Data Analysis

The WT and GPR41KD BRECs were seeded in DMEM/F12 media for 24 h each at 37 °C and 5% CO_2_. TRIzol was employed to extract the total RNA from the BRECs (Invitrogen, Shanghai, China). A DS-11 spectrophotometer (DeNovix, Wilmington, DE, USA) was used to confirm the quantity and purity of the RNA. The library was prepared according to the method of Zhan et al., and the samples were sent to BGI Technology Co., Ltd. (Shenzhen, China) for transcriptome sequencing [13].

### 4.4. qRT-PCR

The WT and the GPR41KD BRECs cells were cultured in six-well plates at 5 × 10^4^ cells per well and cultured in DMEM/F12 medium at 37 °C and 5% CO_2_. After 24 h of incubation, the total RNA was extracted from the cultured cells according to the protocol of kit DP451 (Tiangen, Beijing, China). The concentration and quality of the RNA was assessed using a microspectrophotometer (Thermo Scientific, Shanghai, China). Next, 1 μg of RNA was converted to cDNA, according to the protocol of FastKing one-step RT-PCR kit KR123 (Tiangen, Beijing, China). The cDNA was diluted with ddH_2_O to a final concentration of 50 ng/μL. The qRT-PCR primers used are listed in Table 2. The relative level of the target gene was calculated using the 2^−ΔΔCT^ method, using the GAPDH gene as the domain gene.

### 4.5. Western Blot

The Western blot protocol and reagent sources were the same as in the previous experiment [13,52]. The following main antibodies were obtained from Cell Signaling Technology (CST, Shanghai, China): GAPDH, PIK3, AKT, p-AKT, mTOR, p-mTOR, S6K1, p-S6K1, 4EBP1 and p-4EBP1 (1:1000; CST). The horseradish peroxidase (HRP)-conjugated secondary antibodies are goat antirabbit IgG (1: 5000; CST). The target bands were detected using the Pierce ECL Plus Western Blotting Substrate (Thermo Scientific). Band density was quantified using ImageJ software (version 1.48, National Institute of Health, Bethesda, MD, USA).

### 4.6. Statistical Analysis

Using the Kolmogorov-Smirnov and Levene’s tests, respectively, the gathered data were checked to see if they complied with the normal distribution test and the homogeneity of variances test before being subjected to analysis. One-way ANOVA was used for statistical analysis, followed by post hoc multiple comparisons of the treatment means using the statistical software package Statistical Package for the Social Sciences (SPSS) (version 26, IBM, New York, NY, USA). The experiment was split into two groups, and the independent samples *t*-test was used to test the statistical analysis. Significant differences were defined as *p* < 0.05 and extremely significant differences as *p* < 0.01, respectively. All of the data are presented as means ± standard error of the results of three independent experiments.

## 5. Conclusions

In summary, our findings indicated that GPR41KD suppressed cell proliferation. This may be related to the fact that GPR41KD inhibits the levels of genes related to amino acid and glucose transport, altering the cell cycle progression and expression of key proteins of the PIK3-AKT-mTOR signaling pathway.

## Figures and Tables

**Figure 1 ijms-24-04203-f001:**
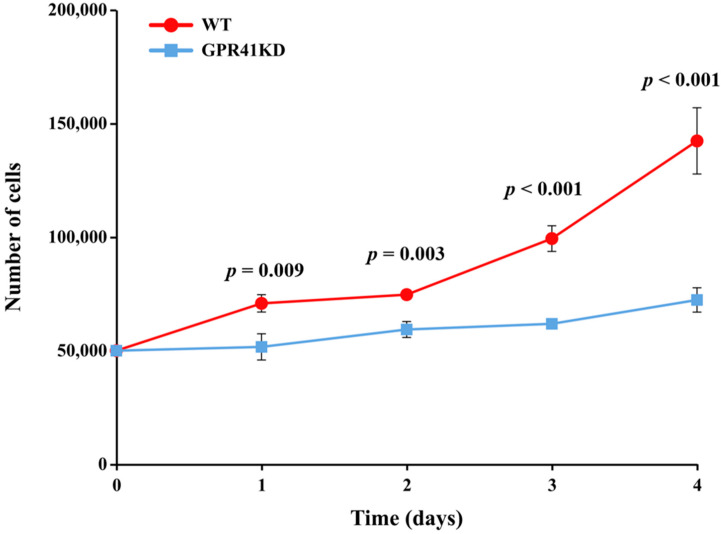
The proliferation of WT and GPR41KD BRECs. These cells were cultured for 1, 2, 3, or 4 days. Data are presented as means ± SEM (*n* = 3).

**Figure 2 ijms-24-04203-f002:**
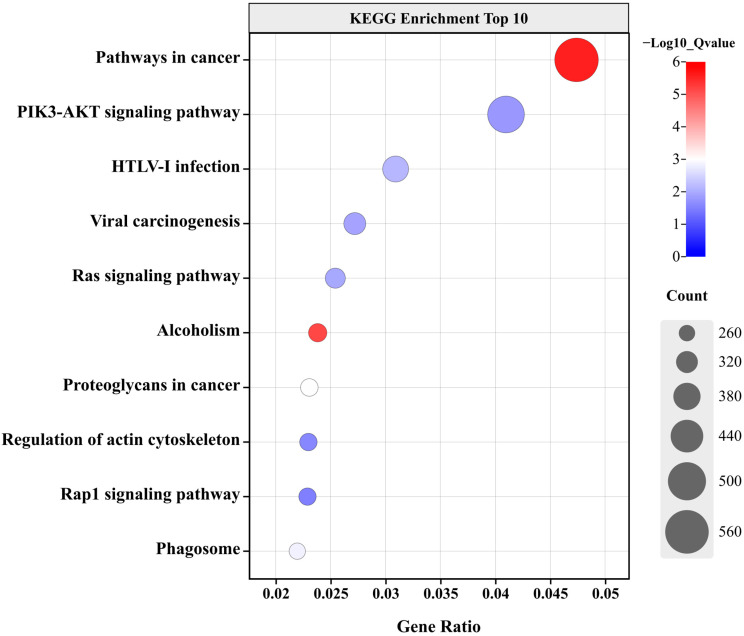
Effect of GPR41 KD on the transcriptome of BRECs (*n* = 3). Results of KEGG enrichment analysis in the GPR41 KD group compared to WT.

**Figure 3 ijms-24-04203-f003:**
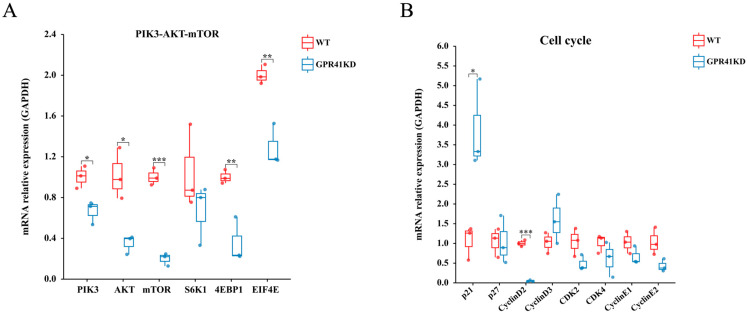
Expression of genes involved in PIK3-AKT-mTOR (**A**) and cell cycle signaling pathway (**B**) regulators in WT and GPR41KD BRECs in vitro (*n* = 3). * *p* ≤ 0.05, ** *p* ≤ 0.01, *** *p* ≤ 0.001.

**Figure 4 ijms-24-04203-f004:**
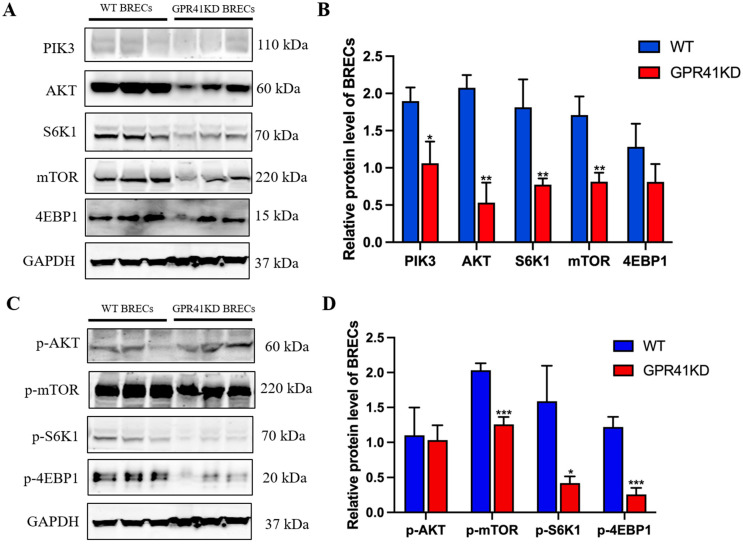
GPR41KD inhibits the expression of key proteins of the PIK3-AKT-mTOR pathway (*n* = 3). (**A**,**B**) GPR41KD inhibited the expression of PIK3, AKT, S6K1, mTOR. (**C**,**D**) GPR41KD inhibited the expression of p-mTOR, p-S6K1, p-4EBP1. GAPDH serve as the loading controls for normalization. The data are expressed as the mean ± SEM and analyzed by *t*-test. * *p* ≤ 0.05, ** *p* ≤ 0.01, *** *p* ≤ 0.001.

**Table 1 ijms-24-04203-t001:** RNA-seq analysis of the effect of GPR41KD on the differential expression of genes related to amino acid transport, glucose transport, PIK3-AKT-mTOR and cell cycle in BRECs.

Gene	Gene Description	Fold Change	*p*-Value
Amino Acid Transport			
SLC1A1	Solute carrier family 1 member 1	0.18	<0.001
SLC1A5	Solute carrier family 1 member 5	0.43	<0.001
SLC7A1	Solute carrier family 7 member 1	0.39	<0.001
SLC7A7	Solute carrier family 7 member 7	0.37	<0.001
SLC7A11	Solute carrier family 7 member 11	0.34	<0.001
SLC38A1	Solute carrier family 38 member 1	0.28	<0.001
Glucose Transport			
SLC2A5	Solute carrier family 2 member 5	0.006	<0.001
SLC2A11	Solute carrier family 2 member 11	2.8	0.02
SLC2A13	Solute carrier family 5 member 3	0.39	0.008
SLC5A3	Solute carrier family 5 member 3	0.27	<0.001
SLC5A9	Solute carrier family 5 member 9	3.6	0.002
PIK3-AKT-mTOR			
PIK3CA	Phosphatidylinositol-4,5-bisphosphate 3-kinase catalytic subunit alpha	0.46	<0.001
PIK3CB	Phosphatidylinositol-4,5-bisphosphate 3-kinase catalytic subunit beta	0.47	<0.001
PIK3CG	Phosphatidylinositol-4,5-bisphosphate 3-kinase catalytic subunit gamma	0.08	<0.001
PIK3C2A	Phosphatidylinositol-4-phosphate 3-kinase catalytic subunit type 2 alpha	0.37	<0.001
PIK3C2B	Phosphatidylinositol-4-phosphate 3-kinase catalytic subunit type 2 beta	0.47	<0.001
mTOR	Mechanistic target of rapamycin kinase	0.34	<0.001
4EBP2	Eukaryotic translation initiation factor 4E binding protein 2	0.46	0.03
RPS6KA2	Ribosomal protein S6 kinase A2	0.12	<0.001
KRAS	KRAS proto-oncogene, GTPase	0.32	<0.001
ERK4	Mitogen-activated protein kinase 4	0.30	<0.001
Cell Cycle			
CCND2	Cyclin D2	0.13	<0.001
CDK6	Cyclin dependent kinase 6	0.32	<0.001
CDC25A	Cell division cycle 25A	0.43	<0.001
MYCL	MYCL proto-oncogene	0.35	<0.001
BCL2	BCL2 apoptosis regulator	0.28	<0.001
CDK18	Cyclin dependent kinase 18	0.50	<0.001
GSK3B	Glycogen synthase kinase 3 beta	0.43	<0.001

**Table 2 ijms-24-04203-t002:** Primers for real-time PCR analyses.

Gene	Primer Sequence, 5′ to 3′ *	Product Size (bp)	Source
GAPDH	F: GGGTCATCATCTCTGCACCT	176	NM_001034034.2
	R: GGTCATAAGTCCCTCCACGA		
p15^INK4B^	F: ACCCGGAAGTCACCTCAATT	226	NM_001075894
	R: GGGGCTCTCTGAATCCTACC		
p16^INK4A^	F: CCTCTGAAGTCAAAAGGCGG	121	XM_010807758
	R: AAATCCTGACTCGTGGTGGG		
p21^Cip1^	F: GCAGACCACATGACAGATT	205	XM_005223326.4
	R: GTATGTACAAGAGAGGCGT		
p27^Kip1^	F: GACCTGCCGCAGATGATTCC	249	NM_001100346.1
	R: CCATTCTTGGAGTCAGCGAT		
Cyclin E1	F: TTGACAGGACTGTGAGAAGC	187	XM_024978361.1
	R: TTCAGTACAGGCAGTGGCGA		
Cyclin E2	F: CTGCATTCTGAGTTGGAACC	229	XM_025001684.1
	R: CTTGGAGCTTAGGAGCGTAG		
CDK4	F: ACTCTGTATCGTGCTCCAGAAG	114	XM_005206553.4
	R: CAGAAGAGAGGCTTTCGAGAA		
Cyclin D1	F: GCACTTCCTCTCCAAGATGC	204	NM_001046273.2
	R: GTCAGGCGGTGATAGGAGAG		
Cyclin D2	F: CCAGACCTTCATCGCTCTGT	163	XM_024992177.1
	R: GATCTTTGCCAGGAGATCCA		
PIK3	F: GATGCTACCTTACGGCTGCT	215	NM_001206047
	R: CGGCACAGGATAGGGTAAAC		
AKT	F: CACCATTACGCCACCTGAC	233	NM_173986
	R: CACTCAAACGCATCCAGAAA		
mTOR	F: ATGCTGTCCCTGGTCCTTATG	178	XM_002694043
	R: GGGTCAGAGAGTGGCCTTCAA		
S6K1	F: ATGAAAGCATGGACCATGGG	199	NM_205816
	R: CCGGTATTTGCTCCTGTTAC		
4EBP1	F: GAACTCACCTGTGACCAAGA	157	NM_001077893
	R: CTCAAACTGTGACTCTTCACC		
EIF4E	F: GAAGACTTTTGGGCTCTGTAC	250	NM_174310
	R: CAGCTCCACATACATCATCA		

* F, forward; R, reverse.

## Data Availability

The datasets generated during the current study are available from the corresponding author on reasonable request.

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
