# Peer review of "GPR41 Regulates the Proliferation of BRECs via the PIK3-AKT-mTOR Pathway"

_ijms, 2023, doi:10.3390/ijms24044203_

Round 1
Reviewer 1 Report
The manuscript submitted by Meng, Z and co-workers addresses how the G-protein-coupled receptor 41 (GPR41) regulate proliferation of bovine rumen epithelial cells (BRECs). They found that knockdown of GPR41 in BRECs leads to a reduced proliferation and, mechanistically, they argue that the effect is mediated by PI3K-AKT-mTOR pathway. The manuscript writing can be improved (e.g. some abbreviations appear before the definition of the word, the word “appertained” appears three times on the same page…)
SCFAs are important signalling molecules derived from the rumen microbiota that regulate a variety of physiological functions of the rumen including immunity and barrier effect. SCFAs play a pivotal effect in regulating the proliferation and development of BRECs though GPR41, which works as a receptor for SCFAs, but the signalling pathways involved in the process are not fully understood. Although it is biologically relevant to understand the mechanisms that control BRECs development as this may lead to an increase in ruminant productivity, this reviewer considers that more data are needed to support the actual claims of the paper.
Meng, Z et al. shows 3 figures with a single panel each. This reviewer considers that a more comprehensive analysis is necessary to publish a paper in this journal and misses the presence of alternative methods to validate the hits from the RNAseq.
Some other comments about the figures are:
Figure 1 shows a growth curve of WT and GPR41KD BRECs. I would appreciate to see the knowdown efficiency preferably over time. Also, more experiments would be required to understand if GPR41KD affects proliferation, cell death, apoptosis. Moreover, as they find some of the cell cycle regulators being differentially expressed, they could analyse whether the cell cycle progression is affected upon GPR41 KD.
Figure 2 is a heat map including the main findings of the RNAseq. Colour code as well as order of samples (WT vs KD) is quite confusing. I consider that, nowadays, a clearer way of presenting RNAseq data should be used.
Figure 3 contains immunoblots showing the effect of GPR41KD in the protein expression of several components of the PI3K-AKT-mTOR pathway and their phosphorylation. The general quality of the blots is susceptible to be improved: some bands are not clear enough (eg. PI3K blot), molecular size markers should be included in every panel, unspecific bands should be indicated. In addition to Western Blot, it could be helpful to include qPCR data of these and other markers as well as downstream effectors of the pathway. Another relevant question for this reviewer is to test whether the treatment with AKT/mTOR inhibitors phenocopies the functional results of GPR41KD. It would be also helpful to show the ratio phospho-protein/total-protein for each of the analysed proteins.
Author Response
Dear Reviewer,
On behalf of my co-authors, we thank you very much for giving us an opportunity to revise our manuscript, and we also appreciate reviewers very much for their positive and constructive comments and suggestions on our manuscript entitled “GPR41 regulates the proliferation of BRECs via the PIK3-AKT-mTOR pathway” (Manuscript ID: ijms-2105976).
We revised the manuscript according to these comments and suggestions. In general, we have tried our best to revise our manuscript and provide the point-by-point responses. All changes were marked in red using the “Track Changes” function in the revised manuscript. Attached please find our responses to the referees’ comments.
The following is a summary list of changes:
(1) The manuscript submitted by Meng, Z and co-workers addresses how the G-protein-coupled receptor 41 (GPR41) regulate proliferation of bovine rumen epithelial cells (BRECs). They found that knockdown of GPR41 in BRECs leads to a reduced proliferation and, mechanistically, they argue that the effect is mediated by PI3K-AKT-mTOR pathway. The manuscript writing can be improved (e.g. some abbreviations appear before the definition of the word, the word “appertained” appears three times on the same page…)
Response 1: Thank you very much for your question. The language has been strictly reviewed and corrected. We highlighted the modification sentence in yellow color.
(2) SCFAs are important signalling molecules derived from the rumen microbiota that regulate a variety of physiological functions of the rumen including immunity and barrier effect. SCFAs play a pivotal effect in regulating the proliferation and development of BRECs though GPR41, which works as a receptor for SCFAs, but the signalling pathways involved in the process are not fully understood. Although it is biologically relevant to understand the mechanisms that control BRECs development as this may lead to an increase in ruminant productivity, this reviewer considers that more data are needed to support the actual claims of the paper.
Response 2: Thank you very much for your question. We have reorganized the available data and rephrased our claims.
(3) Meng, Z et al. shows 3 figures with a single panel each. This reviewer considers that a more comprehensive analysis is necessary to publish a paper in this journal and misses the presence of alternative methods to validate the hits from the RNAseq.
Response 3: Thank you very much for your question. Based on the previous test images, we have added to them and explained them in detail in the answers below.
(4) Figure 1 shows a growth curve of WT and GPR41KD BRECs. I would appreciate to see the knockdown efficiency preferably over time. Also, more experiments would be required to understand if GPR41KD affects proliferation, cell death, apoptosis. Moreover, as they find some of the cell cycle regulators being differentially expressed, they could analyse whether the cell cycle progression is affected upon GPR41 KD.
Response 4: Thank you very much for your question. The knockout cell lines used in our experiments are from a previous study[1–3] and are based on CRISPR/Cas9-mediated KO, whose knockout efficiency does not change in a short period of time. And we also tried to detect the GPR41 polyclonal antibody (Invitrogen, Cat No: PA5-75521) expression level by Western blot. Unfortunately, no GPR41 band was found. Besides, in our previous study, we have found that GPR41 KD inhibits cell proliferation by disrupting the progression of the G0-G1 and S phases of the cell cycle[2]. Therefore, it is also based on these studies that triggered us to think about whether GPR41 KD affects proliferation through other pathways. And this study was based on transcriptomic results and mainly explored the effect of GPR41KD on cell proliferation through the PIK3-AKT-mTOR signaling pathway.
- Zhan, K.; Gong, X.; Chen, Y.; Jiang, M.; Yang, T.; Zhao, G. Short-Chain Fatty Acids Regulate the Immune Responses via G Protein-Coupled Receptor 41 in Bovine Rumen Epithelial Cells. Front. Immunol. 2019, 10, 2042, doi:10.3389/fimmu.2019.02042.
- Yang, T.; Zhan, K.; Ning, L.; Jiang, M.; Zhao, G. Short‐chain Fatty Acids Inhibit Bovine Rumen Epithelial Cells Proliferation via Upregulation of Cyclin‐dependent Kinase Inhibitors 1A, but Not Mediated by G Protein‐coupled Receptor 41. J Anim Physiol Anim Nutr. 2020, 104, 409–417, doi:10.1111/jpn.13266.
- Yang, T.; Datsomor, O.; Jiang, M.; Ma, X.; Zhao, G.; Zhan, K. Protective Roles of Sodium Butyrate in Lipopolysaccharide-Induced Bovine Ruminal Epithelial Cells by Activating G Protein-Coupled Receptors 41. Front. Nutr. 2022, 9, 842634, doi:10.3389/fnut.2022.842634.
(5) Figure 2 is a heat map including the main findings of the RNAseq. Colour code as well as order of samples (WT vs KD) is quite confusing. I consider that, nowadays, a clearer way of presenting RNAseq data should be used.
Response 5: Thank you very much for your question. We redraw the picture of the transcriptome results, The recreated images are detailed in Figure 2 in the original article.
(6) Figure 3 contains immunoblots showing the effect of GPR41KD in the protein expression of several components of the PI3K-AKT-mTOR pathway and their phosphorylation. The general quality of the blots is susceptible to be improved: some bands are not clear enough (eg. PI3K blot), molecular size markers should be included in every panel, unspecific bands should be indicated. In addition to Western Blot, it could be helpful to include qPCR data of these and other markers as well as downstream effectors of the pathway. Another relevant question for this reviewer is to test whether the treatment with AKT/mTOR inhibitors phenocopies the functional results of GPR41KD. It would be also helpful to show the ratio phospho-protein/total-protein for each of the analysed proteins.
Response 6: Thank you very much for your question. We apologize for the lack of clarity in the quality of our blots, but the original images of all strips are represented in the supplemental data. The antibody PIK3 antigen is derived from a synthetic peptide corresponding to human PI3K p110γ and may have low cross-reactivity with bovine. We redrew the images of the western blot results and labeled the molecular size of the proteins. The RNA-seq and qPCR data for markers (mainly PI3K-AKT-mTOR pathway key proteins) have been shown in Tables 2 and 3. Thank you very much for your suggestion that treatment with AKT/mTOR inhibitors is an excellent way to explore their effect on proliferation. However, for this study, the only variable was GPR41KD, which we therefore believe is the main cause of proliferation inhibition, cell cycle alterations, and changes in the PI3K-AKT-mTOR pathway. Our emphasis is on the changes in the PI3K-AKT-mTOR pathway due to GPR41KD. Finally, integrating the suggestions of reviewer 2, each analyzed protein is shown after correction by GAPDH.
Once again, thank you very much for your comments and suggestions. And we hope that the revised manuscript can be accepted by International Journal of Molecular Sciences. If further revision is necessary, please contact me at: zulutango7@163.com
Thank you and best regards.
Sincerely yours,

Reviewer 2 Report
The results of this research showed that GPR41 may affect the proliferation of BRECs by the PIK3-AKT-mTOR signalling pathway. At present, several contents need to be modified.
(1) The full name corresponding to the abbreviation should be displayed where it first appears. For example, Lines 14 and 15.
(2) Line 22-25 : Which proteins belong to IK3-AKT-mTOR signalling pathway ? It is recommended that they be described in the appropriate locations.
(3)Line 69: Author mentioned that the BRECs proliferated robustly within 4 d of culture post-seeding. But proliferation in GPR41KD BRECs is not so obvious. So this sentence needs to be revised.
(4)The results of western blot need to be supplemented by a gray value ( target protein / GAPDH ) significance test analysis.
(5) In the discussion section, the similarities and differences between the results of this study and the 10th reference (Yang et al, 2020) need to be discussed.
(6) Language needs to be further refined by professional companies or English-speaking experts.
Author Response
Dear Reviewer,
On behalf of my co-authors, we thank you very much for giving us an opportunity to revise our manuscript, and we also appreciate reviewers very much for their positive and constructive comments and suggestions on our manuscript entitled “GPR41 regulates the proliferation of BRECs via the PIK3-AKT-mTOR pathway” (Manuscript ID: ijms-2105976).
We revised the manuscript according to these comments and suggestions. In general, we have tried our best to revise our manuscript and provide the point-by-point responses. All changes were marked in red using the “Track Changes” function in the revised manuscript. Attached please find our responses to the referees’ comments.
The following is a summary list of changes:
(1) The full name corresponding to the abbreviation should be displayed where it first appears. For example, Lines 14 and 15.
Response 1: Thank you very much for your question. The full names and abbreviations have been strictly reviewed and corrected. And some other gene names have been described in Table 2. We highlighted the modification sentence in yellow color.
(2) Line 22-25: Which proteins belong to PIK3-AKT-mTOR signalling pathway? It is recommended that they be described in the appropriate locations.
Response 2: Thank you very much for your question. PIK3, AKT,4EBP1 and mTOR are key proteins of the PIK3-AKT-mTOR signaling pathway. This sentence has been revised as “It was evident that GPR41KD BRECs downregulated the level of PIK3-Protein kinase B (AKT)-mammalian target of rapamycin (mTOR) signaling pathway core proteins, such as PIK3, AKT, eukaryotic translation initiation factor 4E binding protein 1 (4EBP1) and mTOR contrasted with the WT cells.” We highlighted the modification sentence in yellow color.
(3) Line 69: Author mentioned that the BRECs proliferated robustly within 4 d of culture post-seeding. But proliferation in GPR41KD BRECs is not so obvious. So this sentence needs to be revised.
Response 3: Thank you very much for your question. This sentence has been revised as “In WT groups, cell proliferation increased steadily from day 1 to day 4”. We highlighted the modification sentence in yellow color.
(4) The results of western blot need to be supplemented by a gray value (target protein / GAPDH ) significance test analysis.
Response 4: Thank you very much for your question. We have re-analyzed the western blot results and changed the images. The recreated images are detailed in Figure 3 in the original article.
(5) In the discussion section, the similarities and differences between the results of this study and the 10th reference (Yang et al, 2020) need to be discussed.
Response 5: Thank you very much for your question. The findings of Yang et al. are the basis of our study. Yang et al. explained the cell proliferation inhibition caused by GPR41KD mainly in terms of cell cycle. Based on this result, we used the same knockout cell lines to localize to the PIK3-AKT-mTOR signaling pathway by transcriptomic results, which was of interest to us. We have reworked the discussion and highlighted the trim sentence in yellow.
(6) Language needs to be further refined by professional companies or English-speaking experts.
Response 6: Thank you very much for your question. The language has been strictly reviewed and corrected. We highlighted the modification sentence in yellow color.
Once again, thank you very much for your comments and suggestions. And we hope that the revised manuscript can be accepted by International Journal of Molecular Sciences. If further revision is necessary, please contact me at: zulutango7@163.com
Thank you and best regards.
Sincerely yours,

Round 2
Reviewer 1 Report
The reviewed manuscript addresses some of the concerns raised by this reviewer. Nonetheless, I still consider that the manuscript should be improved in several aspects:
The hypothesis of the paper is not clearly stated (lines 58-64).
Table 1 should be sent to Supplementary or be included somewhere else, as it is part of the methods and not the results.
Table 2 and 3 are rather confusing as it is not well described where the data come from. Table 2 is somehow redundant with Figure 2. On the other hand, qPCR data included in Table 3 would be more informative if represented as a graph showing the replicates and standard deviation.
The colour code in Figure 2 is not helpful as red and blue are used to distinguish between WT and KD as well as for the heat map.
The figure legend in Figure 3 is incomplete. It should clarify what the bar charts are based on as well as the number and type (experimental, biological...) of replicates included.
In general, the manuscript has been improved and the effect of GPR41KD in proliferation and gene expression are clearer. However, more mechanistic insight would be needed to prove that the observed phenotype is a direct consequence of the molecular changes.
Author Response
Dear Reviewer,
On behalf of my co-authors, we thank you very much for giving us an opportunity to revise our manuscript, and we also appreciate reviewers very much for their positive and constructive comments and suggestions on our manuscript entitled “GPR41 regulates the proliferation of BRECs via the PIK3-AKT-mTOR pathway” (Manuscript ID: ijms-2105976).
We revised the manuscript according to these comments and suggestions. In general, we have tried our best to revise our manuscript and provide the point-by-point responses. All changes were marked in red using the “Track Changes” function in the revised manuscript. Attached please find our responses to the referees’ comments.
The following is a summary list of changes:
(1) The hypothesis of the paper is not clearly stated (lines 58-64).
Response: Thank you very much for your question. We have modified the hypothesis. The revised parts of the manuscript are highlighted in red color.
(2) Table 1 should be sent to Supplementary or be included somewhere else, as it is part of the methods and not the results.
Response: Thank you very much for your suggestion. We have adjusted the questions in Table 1 and placed them in the Materials and Methods section. We highlighted the modification sentence in red color.
(3) Table 2 and 3 are rather confusing as it is not well described where the data come from. Table 2 is somehow redundant with Figure 2. On the other hand, qPCR data included in Table 3 would be more informative if represented as a graph showing the replicates and standard deviation.
Response: Thank you very much for your questions and suggestions. We have modified Tables 2 and 3 and highlighted the modification sentence in red color. We have plotted the data changes in Table 3 as a picture, and the details of the changed picture are shown in Figure 3 in the original article.
(4) The colour code in Figure 2 is not helpful as red and blue are used to distinguish between WT and KD as well as for the heat map.
Response: Thank you very much for your question. We have reworked the Figure 2.
(5) The figure legend in Figure 3 is incomplete. It should clarify what the bar charts are based on as well as the number and type (experimental, biological...) of replicates included.
Response: Thank you very much for your question. We have refined the legend and highlighted the modification sentence in red color.
Once again, thank you very much for your comments and suggestions. And we hope that the revised manuscript can be accepted by International Journal of Molecular Sciences. If further revision is necessary, please contact me at: zulutango7@163.com
Thank you and best regards.
Sincerely yours,